# An Empirical Comparison of Portuguese and Multilingual BERT Models for Auto-Classification of NCM Codes in International Trade

Roberta Rodrigues de Lima [1,*], Anita M. R. Fernandes [1,*], James Roberto Bombasar [2], Bruno Alves da Silva [1], Paul Crocker [3] and Valderi Reis Quietinho Leithardt [4,5]

1 Laboratory of Applied Intelligence, School of the Sea Science and Technology-University of Vale do Itajaí, Itajaí 88302-901, Brazil; silvabruno@edu.univali.br
2 Analysis and Systems Development Course, Centro Universitário Avantis, Balneário Camboriú 88339-125, Brazil; james.bombasar@uniavan.edu.br
3 Instituto de Telecomunicações e Departamento de Informática, Universidade da Beira Interior, 6201-001 Covilhã, Portugal; crocker@di.ubi.pt
4 COPELABS, Lusófona University of Humanities and Technologies, Campo Grande 376, 1749-024 Lisboa, Portugal; valderi@ipportalegre.pt
5 VALORIZA, Research Center for Endogenous Resources Valorization, Instituto Politécnico de Portalegre, 7300-555 Portalegre, Portugal
* Correspondence: robertalima@edu.univali.br (R.R.d.L.); anita.fernandes@univali.br (A.M.R.F.)

**Abstract:** Classification problems are common activities in many different domains and supervised learning algorithms have shown great promise in these areas. The classification of goods in international trade in Brazil represents a real challenge due to the complexity involved in assigning the correct category codes to a good, especially considering the tax penalties and legal implications of a misclassification. This work focuses on the training process of a classifier based on bidirectional encoder representations from transformers (BERT) for tax classification of goods with MCN codes which are the official classification system for import and export products in Brazil. In particular, this article presents results from using a specific Portuguese-language-pretrained BERT model, as well as results from using a multilingual-pretrained BERT model. Experimental results show that Portuguese model had a slightly better performance than the multilingual model, achieving an MCC 0.8491, and confirms that the classifiers could be used to improve specialists' performance in the classification of goods.

**Keywords:** NCM classification; natural language processing; multilingual BERT; Portuguese BERT; transformers; NLP; BERT

## 1. Introduction

The Mercosur Common Nomenclature (MCN or NCM) is a system used by the South American trade bloc Mercosur to categorize goods in international trade and to facilitate customs control [1]. The MCN is divided into 96 parts called "chapters". These contain more than 10,000 unique MCN codes. An MCN code is an eight-digit numeric code than represents the goods and is required in the process of importing products in Brazil.

The process of classifying goods can constitute a real challenge due to the complexity involved in assigning the right code to each imported good given the substantial number of codes and the technical details involved in their specification. During the import process, one of the first documents required by Brazil is the Import Declaration in which the MCN code must be assigned to the product. In the case of a missing document or a misclassification of the MCN Code, the fines can be significant—thereby making classification a key challenge.

Since the proposition of the transformer model in [2] the natural language processing (NLP) area has been hugely impacted by this model that does not need recurrences layers and is only based on attention mechanisms. The bidirectional encoder representations from transformers (BERT) model was proposed two years later by [3] stacking only encoder layers from the transformers and achieving state of the art results in 11 natural language processing (NLP) tasks in the GLUE Benchmark thus allowing many NLP tasks to take advantage of this approach. Due to its transfer learning process, BERT models allow for less data and computing time to fine tune a model for a specific task, enhancing and facilitating its use for different purposes.

The Brazilian Revenue Service currently maintains Brazil's international trade data in website called Siscori. This website contains all data relative to Brazilian imports and exports, including a detailed description of the goods and their respective NCM Code. The focus of this work is to use this international trade data to fine-tune a classifier in order to develop a tool that could improve foreign trade analysts' performance when dealing with the classification of goods process. The multilingual BERT model proposed in [3] and the Portuguese BERT (BERTimbau) proposed in [4] will be used. In this article, an assessment of these models' abilities to successfully categorize and attribute NCM codes will be made, as well as making an empirical comparison between the performance of the two models.

## 2. Materials and Methods

### 2.1. Mercosur Common Nomenclature

The Mercosur Common Nomenclature (NCM) was created with the aim of standardizing the classification of goods and to facilitate customs control among the countries that belong to the Mercosur: Argentina, Brazil, Paraguay, Uruguay, and Venezuela [1]. The NCM is based on the harmonized system (HS) which is maintained by the World Customs Organization (WCO) [5]. The 8-digit code is composed by the Harmonized System 6-digit code extended by two additional codes as illustrated in Figure 1.

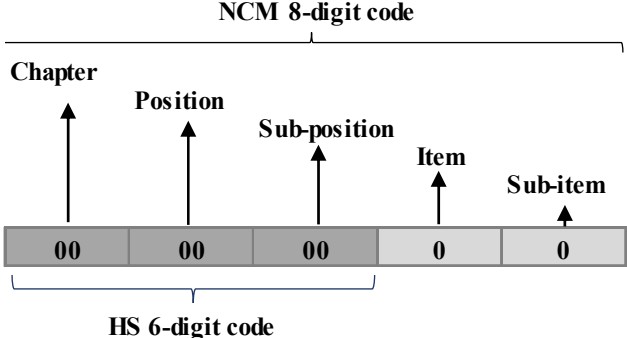

**Figure 1.** Composition of the NCM Code.

The composition of the MCN code starts with the first two digits from the HS code that specify the chapter followed by two digits that refer to the position. The next two codes are related to the sub-position and the last two digits represents the item and the sub-item. The addition of the last two digits allows for a more detailed specification of the goods based on each country's specific needs [1].

### 2.2. Tax Classification of Goods

The main use for the MCN codes in Brazil is for the tax classification of goods. Fees and taxes will be applied in the import process according to the code assigned to the good. According to the Brazilian Revenue Service, the MCN is also used in customs valuation, in statistical data involving import and export data and in import licenses, for special customs regimes such as goods identification.

The classification of goods, which is the focus of this work, is the process of assigning an MCN code to the good according to its technical features and characteristics. The MCN

codes are maintained in tables provided by the Brazilian Revenue Service which aggregate more than 10,000 codes distributed in 21 sections and 96 chapters.

The chapters start with a chapter for live animals and end with a chapter for art works such as paintings [5]. An example is shown in Table 1, Chapter 90 from [6], which is related to photographic and cinematography products.

**Table 1.** Excerpt from Chapter 90 in MCN table [6].

| MCN | DESCRIPTION | CET (%) |
|---|---|---|
| 90.10 | Aparelhos e material dos tipos usados nos laboratórios fotográficos ou cinematográficos, não especificados nem compreendidos noutras posições do presente Capítulo; negatoscópios; telas para projeção. | |
| 9010.10 | -Aparelhos e material para revelação automática de filmes fotográficos, de filmes cinematográficos ou de papel fotográfico, em rolos, ou para copiagem automática de filmes revelados em rolos de papel fotográfico | |
| 9010.10.10 | Cubas e cubetas, de operação automática e programáveis | 0BK |
| 9010.10.20 | Ampliadoras-copiadoras automáticas para papel fotográfico, com capacidade superior a 1.000 cópias por hora | 0BK |
| 9010.10.90 | Outros | 14BK |
| 9010.50 | -Outros aparelhos e material para laboratórios fotográficos ou cinematográficos; negatoscópios | |
| 9010.50.10 | Processadores fotográficos para o tratamento eletrônico de imagens, mesmo com saída digital | 0BK |
| 9010.50.20 | Aparelhos para revelação automática de chapas de fotopolímeros com suporte metálico | 0BK |
| 9010.50.90 | Outros | 18 |
| 9010.60.00 | -Telas para projeção | 18 |
| 9010.90 | -Partes e acessórios | |
| 9010.90.10 | De aparelhos ou material da subposição 9010.10 ou do item 9010.50.10 | 14BK |
| 9010.90.90 | Outros | 16 |

The main purpose of MCN classification is the collection of import taxes, which are based on the Mercosur's Common External Tariff (CET), as well as the establishment of commercial defense rights, such as in the case of anti-dumping [7]. Thus, the correct identification of the MCN code in the import declaration (ID) is necessary to guarantee the correct collection of taxes, as well as the surcharges that guarantee commercial defense. The MCN code is necessary in the registration of the import license, a mandatory document in the import process in Brazil. According to Brazilian Law 6.759/2009 Art. 706, which regulates the administration of customs activities and the inspection, control, and taxation of foreign trade operations, a 30% fine on the imported good price can be applied if the license is required and not presented during the import process. In accordance with Brazilian Law 6.759/2009 Art. 711, a 1% fine is applied on the customs value of the goods if there is an incorrect classification of the MCN code. In addition, Brazilian Law 9430/1996 Art. 44 states that fines of up to 75% will be applied on the total or difference of tax or contribution in cases of lack of payment or tax collection, lack of declaration, and in cases of inaccurate declaration.

*2.3. Data Classification*

The process of data classification involves assigning categories to each of the objects or entities involved, also known as classes. Reference [8] defines data classification process can be mathematically:

For a given matrix $D$ of training of size $n \times d$ and classes with values between $1, \ldots, k$, associate each of the $n$ rows in $D$, create a training model $M$ that can be used to predict the class of a dimension record $d$ where the record $\gamma \notin D$.

According to [9], classification problems are one of the most common applications in data mining and these tasks are also quite frequent in everyday life.

Classification problems are said to be supervised when the relationship of training data with the class itself is learned [8]. Thus, after the training, a classifier will serve to classify new records in those predefined classes, whose learning took place based on the labeled training data. Finally, the new records that will be provided to the classifier for class determination are called test data and are used to measure the classifier's performance on unknown records.

### 2.4. BERT

The BERT model proposed by [3] is based on a bidirectional multilayer transformer encoder derived from the original implementation of transformers proposed by [2]. BERT works by pre-training deep bidirectional representations from unlabeled data in both context directions. Thus, once pre-trained, a fine-tuning procedure can be performed on top of the model by adding a layer, thereby allowing its use in a wide variety of tasks [3].

According to [3], the procedure with BERT is composed of two steps: a pre-training step followed by a fine-tuning step. In the pre-training stage, the model is trained in unlabeled databases in two large tasks called masked language model (MLM) and next sentence prediction (NSP). At the end, in the fine-tuning step, the model is first initialized with the pre-training parameters, and then later has its weights updated based on the training using labeled data for the specific task. The BERT models provided by the authors are in two size variations: $BERT_{BASE}$ with 12 layers, 768 hidden states, 12 self-attention heads, and a total of 110 million parameters; while BERTLARGE has 24 layers, 1024 hidden states, 12 self-attention heads, and 340 million parameters.

The pre-training stage for the original English BERT model used BooksCorpus with 800 million words according to [3], as well as Wikipedia in English (2500 million words). The authors reiterate the importance of using a document-level basis so that it is possible to extract long continuous sentences in learning. Regarding the embeddings, for BERT, the WordPiece embeddings presented in [10] were used, which had a total vocabulary of 30,000 tokens.

The embeddings used in BERT are built from three vectors: the token embeddings (which are the pre-trained embeddings, related to WordPiece). Next, there are the segment embeddings, which reflect the number of the phrase encoded in a vector, and finally, the position embeddings that bring the position of the word in the phrase. Figure 2, adapted from [3], illustrates the embeddings composition in a simplified way due to WordPiece's use pieces of words, wherein the original words are often broken into smaller terms.

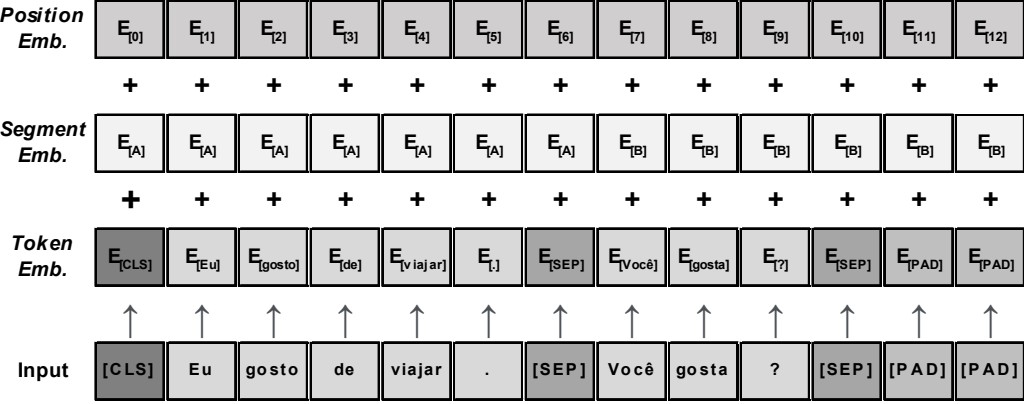

**Figure 2.** Embedding composition.

At this point, it is worth emphasizing that segment and position embeddings are necessary so that it is possible to maintain the order of the input data, since the process is parallelized, allowing simultaneous processing, but maintaining knowledge of this order. Once this understanding regarding embeddings is defined, the two tasks used in BERT training, mentioned above, are taken into account.

In the first one, which corresponds to the masked language model (MLM) step, 15% of these input tokens are randomly masked, and the task to which the model is submitted is precisely this: to predict these tokens, such as can be seen in Figure 3. Of those 15% of tokens that will change, 80% of those will actually be replaced by the mask token, 10% of the time they will be replaced by a completely random token, and the last 10% will have an unchanged token.

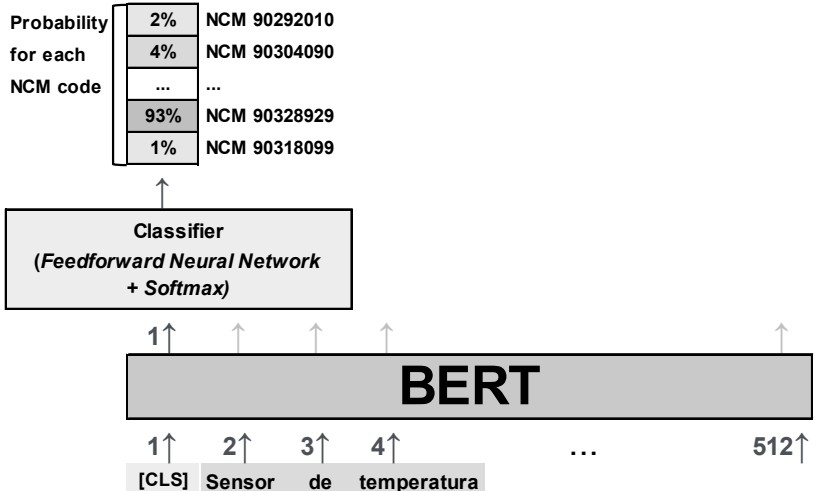

**Figure 3.** Fine-tuning process.

The next sentence prediction (NSP) task is relevant since many downstream tasks such as the question answering task need and are based on an understanding of the relationship between two sentences, which cannot be captured in the previous MLM step [3]. Thus, in this task, the model receives two distinct sentences A and B and must predict whether sentence B is a correct sequence for sentence A or not: in this case, 50% of the examples presented are correct sequences and 50% are random sentences from the base.

The fine-tuning step, which is the focus of this work, tends to be much faster and straightforward than the training process: if it is a classification task, for example, there's simply a need to add a classification layer to the pre-trained model, which will result in all parameters being adjusted for the new task. In this case, these tasks are called downstream, as they present themselves as supervised learning tasks that use a pre-trained model or component. Figure 3, adapted from [11], illustrates the fine-tuning process on a BERT model for the classification of MCN codes from product descriptions, which is the focus of this work.

In summary, after the pre-training phase, the model will have the idea of language and context, and after fine tuning it will be provided with the means to solve the specific problem—e.g., a problem of classification. This approach is referred in [3] as knowledge transfer and its relevance has already been proved in the computer vision field and has also brought state-of-the-art results in a series of NLP tasks. It has helped in the dissemination of models that use transformers for a series of applications, as it enables the achievement of results with reduced training time, since it already starts from a pre-trained model as a basis.

One of the main advantages of using BERT is its transfer learning which allows the timely training of a classifier and requires less data than usually needed when training classifiers from scratch compared with other methods. As stated in [12], a transformer model does not have its words processed in order, but with each word processed in relation to all other words in a sentence, where the model could get the full context involved in each word. Since the model is already pre-trained and words are taken into account in their context, BERT models also usually achieve good accuracy and MCC results with less effort than other traditional methods.

On the other hand, the BERT model's main disadvantage is the relatively high computational resources needed to train the model, despite the fact that the fine-tuning time needed is much less than the pre-train time. A further point to consider is that, due to its transformer nature, the BERT model's decision will be a 'black box' and thus it will be unclear to the users why certain decisions are made, this can be a disadvantage compared to other methods. Note however, an intuition and understanding on what the classifier is considering could be inferred using the LIME (local interpretable model-agnostic explanations) presented in [13].

### 2.4.1. Multilingual BERT

Combined with the BERT model originally trained in English, Ref. [3] also makes available the multilingual BERT, a pre-trained model in 102 different languages, which can be used in the same way as the single language model. According to the authors, this model allows the use of several languages, since it is not practically feasible to maintain so many isolated language models. However, a disadvantage of the multi-language model is that, according to its documentation, it can underperform single-language models, particularly in feature-rich languages. In that case, language-specific pre-training is indicated for increased performance.

### 2.4.2. Portuguese BERT

The BERT model trained entirely in Brazilian Portuguese has been nicknamed BERTimbau [4]. According to the authors, the Portuguese model replicates the pre-training procedures with just a few changes and is available in two sizes with the same number of layers and parameters as the original BERT: base and large. The training was based on the brWaC Corpus, web as corpus in Brazilian Portuguese and the model was evaluated in three distinct downstream tasks—sentence textual similarity, recognizing textual entailment, and named entity recognition—outperforming the multilingual model in both sizes' results [4].

### 3. Procedures

The starting point is obtaining the publicly available data provided by the Brazilian Revenue Service which is then run through the 18 scenarios for each model (Portuguese BERT and Multilingual BERT). Some steps were carried out in order to set up the initial data and preprocess it. Figure 4 illustrates each step on data selection, preprocessing, transformation, and finally on model training and validation.

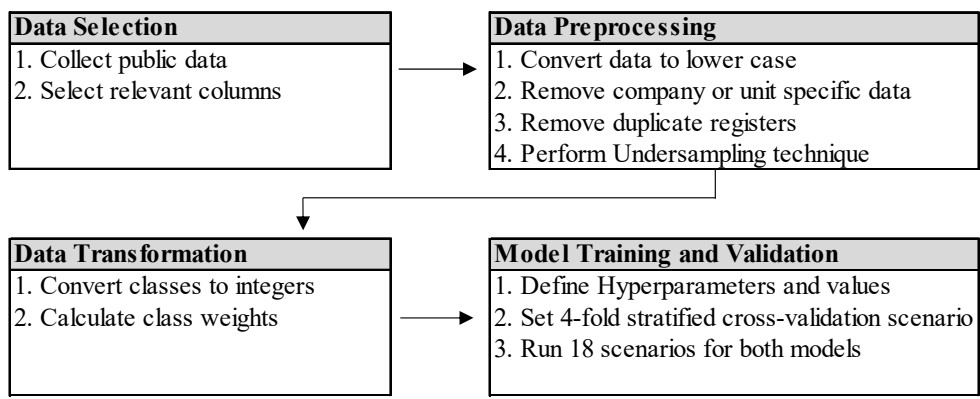

**Figure 4.** Procedures.

### 3.1. Data Selection

The data used in this work was obtained from "Siscori" [14], the website provided by the Brazilian Revenue Service that currently contains all data on the import and export of goods from 2016 to 2021 (Figure 5). The data records include a detailed description on the good (with a maximum of 250 characters), country of origin, country of destination, related

costs, and the MCN code into which the good is classified, as well as other information that was not relevant to this work.

| NCM CODE | PRODUCT DESCRIPTION |
|---|---|
| 90279099 | DISPOSITIVO DE CONTAGEM DE CELULAS DE USO EXCLUSIVO EM EQUIPAMENTO DE ANALISES V(...) |
| 90292010 | 84713545 INDICADOR DE VELOCIDADE E TACOMETRO |
| 90299010 | 37212KWN901 CARCAÇA DO PAINEL DE INSTRUMENTOS, FABRICADO EM PLASTICO E APLICADO (...) |
| 90304090 | 2097251 - EQUIPAMENTO DE TESTE DE CABO IQ CIQ 100 PARA QUALIFICACAO DE REDE DE DADO(...) |
| 90304090 | 5258569 - VERIFICADOR DE CABOS MICROSCANNER POE PARA ETHERNET INDUSTRIAL |
| 90308990 | SENSOR ELETROPENEUMATICO DO SISTEMA DE AR CONDICIONADO DA AERONAVE - LOTE: 190U(...) |
| 90308990 | 503839P - SENSOR DE TEMPERATURA PARA LAVADORA DE ROUPAS DE 24KG DE CAPACIDADE |
| 90308990 | APARELHO PARA MEDIDA DA ISOLAÇÃO ELÉTRICA (RIGIDEZ DIELÉTRICA) DOS CONDUTORES ES(...) |
| 90318099 | SENSOR MAGNETICO(A) PARA MEDIR A VELOCIDADE EM MOTOR DE IGNICAO POR COMPRESSA(...) |
| 90328929 | SENSOR DE TEMPERATURA DA AGUA. **SUFRAMA**SENSOR DE TEMPERATURA DA AGUA, DO CA(...) |
| 90329010 | EEA9794B: PLACA DE CIRCUITO ELETRONICA ULITIZADO NO SISTEMA PALTRONIC, CÓDIGO EEA97(..) |

**Figure 5.** Chapter 90 import data sample [14].

Since this paper makes use of the BERT model to train a classifier that aims to classify descriptions on its respective MCN code, only the product description and MCN code columns were selected for this work. Although the amount of NCM codes is over 10,000 and is currently divided into 96 chapters, the focus of the classifier developed in this paper will be to predict MCN codes inside a single chapter.

Regarding the choice of this chapter, reference [15] presents a classifier to the harmonized system (HS) codes using background nets which focuses on Chapters 22 and 90 since both of these chapters are more prone to classification errors in daily classification tasks made by international business analysts than any other chapter. Reference [15] states that the accuracy for Chapter 90 has shown to be much lower than Chapter 22, which the authors consider to have direct relation with two main factors. The first one is related to short record descriptions that do not contain sufficient information for the model to learn properly and also high level descriptions that frequently refer directly to the HS nomenclature itself, this makes learning difficult when dealing with specific descriptions in new instances.

Reference [16] presents a classifier for NCM codes using the naïve Bayes algorithm and focuses on Chapters 22 and 90 as these are most susceptible to practical errors. Reference [16] shows three scenarios in their work with different difficult levels: the first one on Chapter 22 with data from a single company; the second one also on Chapter 22 but with data from companies randomly selected; and finally, the third one on Chapter 90 containing data also from different companies randomly collected. According to [16] it is also important to take into account that the fact when comparing to the harmonized system, that the NCM system has a deeper hierarchy which leads to the difference in performance between classifiers. In addition to this, according to [16], Portuguese is also a more complex language than English that has more regular and uniform sentences allowing the models prediction to be more accurate. Given that the difference in the amount of NCM codes in Chapter 90 is more than 10 times the amount of NCM codes in Chapter 22, this paper will focus only on Chapter 90 since it tends to have more difficulties involved. As mentioned, the Siscori website provides Brazilian import data from 2016 to 2021, and the import data from January 2019 through August 2021 was selected in this experiment, making a total of 7,645,573 records

### 3.2. Data Pre-Processing

Since the data was obtained from the official sources, there was some noise that needed to be removed in a cleaning process before beginning to train the model. At first, as shown in Figure 3, most of the goods described are in upper case and only a few of them come cased as regular sentences, so initially all records were converted to lower case.

Using regular expressions, production and manufacturing dates were removed, as well as product batches and their abbreviations and numbers, since these data are not relevant

to the distinction between classes and can interfere in the training. Additionally, codes and terms related to billing and part number (PN) were removed, since they are specific codes for each company. In addition, extra white space and some special characters present in the sentences that would not contribute to the learning process were also removed.

After the data cleaning phase records that had duplicate descriptions, this is data which most likely come from the same importing companies, were also removed. After the cleaning and removal of duplicate records, the database was reduced from 7,645,573 records to a total of 3,481,090.

Since the processing of this amount of data would still be significant an undersampling technique was carried out with imblearn's RandomUnderSampler. Given that the dataset was unbalanced, the undersampling process for this research kept a ratio of 1:300 samples between the minority and majority classes. The minority class presented only 4 records so the majority class would present 1200 records giving the chosen ratio with the samples being selected randomly. After the undersampling process, the database was reduced from 3,481,090 to 265,818 records keeping the original number of classes of 325. All the pre-processing steps are summarized in Table 2.

**Table 2.** Data records.

|                                      | No. of Records | No. of Classes |
| ------------------------------------ | -------------- | -------------- |
| Original import data                 | 7,645,573      | 325            |
| After cleaning and duplicate removal | 3,481,090      | 325            |
| After undersampling                  | 265,818        | 325            |

*3.3. Data Transformation*

The classifiers were implemented in Python and the main library used to fine-tune the BERT model was simple transformers, by Thilina Rajapakse [17]. For multiclass classification, the format of training and testing data required by the library states that classes must be provided as integers starting from 0 to n. Therefore, a simple transformation of the MCN codes and adaptation of the column names according to the documentation was performed. The MCN codes and their respective indexes were also stored in a dictionary, for later referral after the prediction process.

For this work, the selected multilingual BERT model proposed by [3] was the uncased version on the base form (12 layers). Regarding Portuguese BERT (BERTimbau), since only the cased version was available it was chosen also on its base form. In this case, it is important to emphasize that, for the BERT model to perform tasks called downstream¸ such as classification, the simple transformer library provides models with a classification layer already implemented on top of it. Besides that, simple transformers allow the model to be supplied with a list of weights to be assigned to each class in the case of unbalanced datasets. As the database contains real Brazilian import data and presents a considerable difference in the amount of some goods in relation to others with respect to MCN codes, it is clearly a highly unbalanced base. To make use of this configuration allowed by simple transformers, the weights were calculated inversely proportional to the number of records per class and provided to the model as a parameter, to be used in the loss calculation during training and evaluation, thus minimizing the effects of the unbalanced dataset.

At this point, it is important to emphasize that some steps are necessary so that the texts and classes are provided in the proper format for a BERT model. The first requirement is that the input texts are converted to tokens using the same tokenizer used to originally train BERT, namely the WordPiece embeddings, defined in [10].

BERT also expects special tokens in the texts: for classification tasks, the token '[CLS]' is needed at the beginning of each sentence, and by default the token of '[SEP]' is needed at the end. In addition, it is important to emphasize that BERT can be supplied with variable length sentences, but these must be spaced or truncated to a fixed length and, for this, '[PAD]' tokens are used, which indicate empty spaces. Finally, it is also necessary to

provide a so-called attention mask, responsible for identifying and separating the tokens from the paddings, consisting of a list with a value of 0 for the tokens of '[PAD]' and value 1 for everything else. All these steps are provided by the simple transformers library implementation and do not need extra work to get this working.

### 3.4. Hyperparameter Tuning

Having defined the parameters referred to the model itself, in addition to parameters such as seed to allow the replication of results later, some hyperparameters used in the BERT model were defined as well. Reference [3] specify that most hyperparameters used in fine tuning are the same as in pre-training, except for batch size, learning rate, and number of training epochs. Along these lines, the authors suggest some values that, based on the experiments, have shown to work well for different types of tasks. Reference [3] suggests values for batch size as 16 or 32, for learning rate as $5 \times 10^{-5}, 3 \times 10^{-5}, 2 \times 10^{-5}$, as well as training epochs of 2, 3, and 4. The authors also reiterate that, during their experiments, it was noticed that, for large databases (+100,000 labeled data), the model was far less sensitive to the changes of these parameters than on smaller databases. Because the fine-tuning process is relatively fast, Ref. [3] also recommends carrying out an exhaustive search among these parameters in order to find those that best fit the model. Therefore, a grid search was performed on the training of the classifier of Chapter 90 comprising all 18 scenarios for each BERT model (multilingual and Portuguese).

### 3.5. Hardware and Platform

Since transformers allow parallelization their use with the proper hardware is very important. For this work, the code was developed in the Python language and was executed in a notebook on Google Colab, an interactive environment that is easy to configure and share. For the execution of the experiments used in this research, Google Colab Pro version was used due to the availability of fast GPUs and large amounts of RAM. In this case, all 36 scenarios in this experiment were run on Google Colab Pro and a Tesla P100 PCIe 16 GB GPU was assigned for the notebook.

### 3.6. Model Training and Validation

In order to evaluate the classifier, the training and validation processes were performed using k-fold cross-validation. This is illustrated in Figure 6 in a 4-fold cross-validation which was the number of folds used in this paper's experiments. In [9], it is stated that in k-fold cross-validation where the database comprises N instances then these should be divided into k equal parts, where k is usually a small number, such as 5 or 10. With the database separated into parts a series of k executions is performed using one of the k parts as a test base and the other (k − 1) parts as a training base.

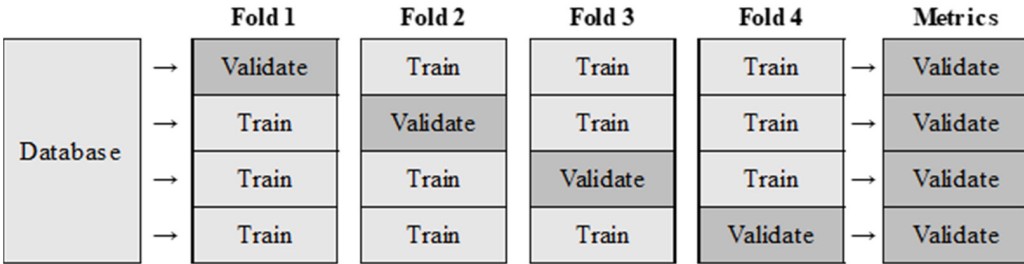

**Figure 6.** Four-fold cross-validation.

Thus, after k executions, reference [9] defines that the total number of correctly classified instances is divided by the total number of instances N, resulting in an overall accuracy of the model. One of the variations of cross-validation is the stratified cross-validation, which—according to [8]—uses a representation proportional to each class in the different folds and generally provides less pessimistic results. As the database used is quite

unbalanced, the stratified cross-validation was run in all experiments with the different hyperparameters to ensure more consistent results. Other scenarios and applications described can be found in [18].

### 3.7. Metrics Selection

To analyze the performance of the classifier, three different metrics were used, as some are more adequate to the characteristics of the database used in this work: accuracy, cross-entropy loss, and Matthews correlation coefficient (MCC).

#### 3.7.1. Accuracy

Accuracy represents the number of instances correctly classified in relation to the total number of instances evaluated. According to [19], although accuracy has been shown to be the simplest and most widespread measure in the scientific literature, it presents some problems in cases in which the performance of unbalanced databases is evaluated. According to the authors, there is an accuracy problem in not being able to distinguish well between different distributions of incorrect classifications.

#### 3.7.2. Cross-Entropy Loss

The cross-entropy loss configures a loss function commonly used in classification tasks, being a performance measure of models, whose output involves probability values. In this case, the cross-entropy loss increases as the predicted probability diverges from the real one, and therefore the objective is for its value to be as close to zero as possible. To calculate the cross-entropy loss in the multiclass case, according to [20] the expression can be written for each instance, where ($k$) has the value 0 or 1, indicating whether the class $k$ is the correct classification for the prediction $\hat{y}^{(k)}$, in the form

$$L(\hat{y}, y) = -\sum_{k}^{K} y^{(k)} \log \hat{y}^{(k)}$$

Class-balanced loss introduces a weight factor that is inversely proportional to the number of instances of a class and is used precisely to address the problem of unbalanced databases [21]. Thus, as this work deals with this case of unbalance and the simple transformers library allows the supply of a vector of weights for the model, these weights are used as multipliers in the cross-entropy loss function.

#### 3.7.3. Matthews Correlation Coefficient

The Matthews correlation coefficient (*MCC*) is calculated directly from the confusion matrix, and its values range between −1 and +1. A +1 coefficient represents a perfect prediction, 0 an average prediction, and −1 an inverse prediction. For binary classifications, *MCC* comprises the following expression based on the binary confusion matrix values

$$MCC = \frac{TP \times TN - FP \times FN}{\sqrt{(TP + FP)(TP + FN)(TN + FP)(TN + FN)}}$$

Since the focus of this work is a multiclass classification problem, reference [22] presents its generalized form for the multiclass case, in which $C_{kl}$ are the elements of the confusion matrix

$$MCC = \frac{\sum_{klm} C_{kk}C_{lm} - C_{kl}C_{mk}}{\sqrt{\sum_{k}(\sum_{l} C_{kl})\left(\sum_{\substack{l' \\ k' \neq k}} C_{k'l'}\right)} \sqrt{\sum_{k}(\sum_{l} C_{lk})\left(\sum_{\substack{l' \\ k' \neq k}} C_{l'k'}\right)}}$$

Reference [23] refers to the MCC in their work as a performance measure of a multiclass classifier, and point out that, in the most general case, the *MCC* presents a good harmonization between discrimination, consistency, and consistent behaviors with a varied number of classes and unbalanced databases. In addition, reference [19] shows that the behavior of the *MCC* remains consistent both in cases of binary and multiclass classifiers.

In their work evaluating binary classifications, reference [23] points out that due to its mathematical properties, the *MCC* can give better, more reliable score on imbalanced datasets. Reference [23] reiterates the fact that the *MCC* criterion is direct and intuitive regarding its score: for a high score, it means that the classifier correctly predicts most negative classes and most positive classes, regardless of their proportions in the dataset. The work of [23] focuses on binary classification and shows that the main differential and advantages of the *MCC* are that it benefits and penalizes for each class, since a good performance of the classifier will occur when it has a good predictive power for each of the classes together. In this sense, regardless of the number of instances of a class being much lower than another (in the case of unbalanced datasets), the *MCC* maintains its overall performance evaluation power consistently.

## 4. Experimental Results

Both multilingual BERT and Portuguese BERT experiments were carried out on a grid search comprising all 18 scenarios that encompasses the combinations of parameters for batch size, epochs, and learning rate suggested in [3]. A 4-fold stratified cross-validation was performed and the results presented on Tables 2 and 3 for cross-entropy loss, accuracy, and Matthews correlation coefficient are averaged among each fold. Both Tables 2 and 3 are sorted by the highest *MCC* result first, since it is the most suitable metric for the unbalanced database.

**Table 3.** Multilingual BERT results for Chapter 90.

| Batch Size | Epochs | Learning Rate | Cross-Entropy Loss | Accuracy | MCC |
|---|---|---|---|---|---|
| 16 | 4 | $5 \times 10^{-5}$ | 0.7326 | 0.8369 | 0.8362 |
| 16 | 4 | $3 \times 10^{-5}$ | 0.7398 | 0.8341 | 0.8334 |
| 32 | 4 | $5 \times 10^{-5}$ | 0.7339 | 0.8314 | 0.8307 |
| 16 | 3 | $5 \times 10^{-5}$ | 0.7580 | 0.8249 | 0.8242 |
| 16 | 4 | $2 \times 10^{-5}$ | 0.7716 | 0.8246 | 0.8239 |
| 32 | 4 | $3 \times 10^{-5}$ | 0.7780 | 0.8206 | 0.8199 |
| 16 | 3 | $3 \times 10^{-5}$ | 0.7827 | 0.8192 | 0.8185 |
| 32 | 3 | $5 \times 10^{-5}$ | 0.7853 | 0.8152 | 0.8145 |
| 16 | 3 | $2 \times 10^{-5}$ | 0.8360 | 0.8062 | 0.8055 |
| 32 | 4 | $2 \times 10^{-5}$ | 0.8475 | 0.8040 | 0.8032 |
| 16 | 2 | $5 \times 10^{-5}$ | 0.8390 | 0.8020 | 0.8013 |
| 32 | 3 | $3 \times 10^{-5}$ | 0.8475 | 0.8015 | 0.8007 |
| 16 | 2 | $3 \times 10^{-5}$ | 0.8876 | 0.7917 | 0.7909 |
| 32 | 2 | $5 \times 10^{-5}$ | 0.8995 | 0.7866 | 0.7857 |
| 32 | 3 | $2 \times 10^{-5}$ | 0.9497 | 0.7807 | 0.7798 |
| 16 | 2 | $2 \times 10^{-5}$ | 0.9718 | 0.7750 | 0.7741 |
| 32 | 2 | $3 \times 10^{-5}$ | 1.0014 | 0.7665 | 0.7656 |
| 32 | 2 | $2 \times 10^{-5}$ | 1.1658 | 0.7379 | 0.7369 |

The experiments performed by the authors, regarding multilingual BERT, demonstrated that the best result regarding *MCC* metric was the one presented where the batch size hyperparameter was set to 16, learning rate to $5 \times 10^{-5}$ and total epochs of 4. This best combination of hyperparameter resulted on a *MCC* of 0.8362, an accuracy result of 0.8369 and a cross-entropy value of 0.7326 as shown on Table 3.

In the work of [16], for the experiment with Chapter 90 without duplicates—which is the most similar to this paper experiment—the classifier presented an average accuracy of 0.8338. In the same scenario, but with terms in English and considering the harmonized

system (HS) classification, the classifier developed by [15], obtained an average accuracy of 0.8330. The result from the multilingual BERT experiment of 0.8362 outperformed both works with a BERT model that was pretrained in 102 languages. Figure 7 shows the hyperparameters optimization process on multilingual BERT experiment with 18 scenarios using weight and biases in [24] to illustrate different scenarios.

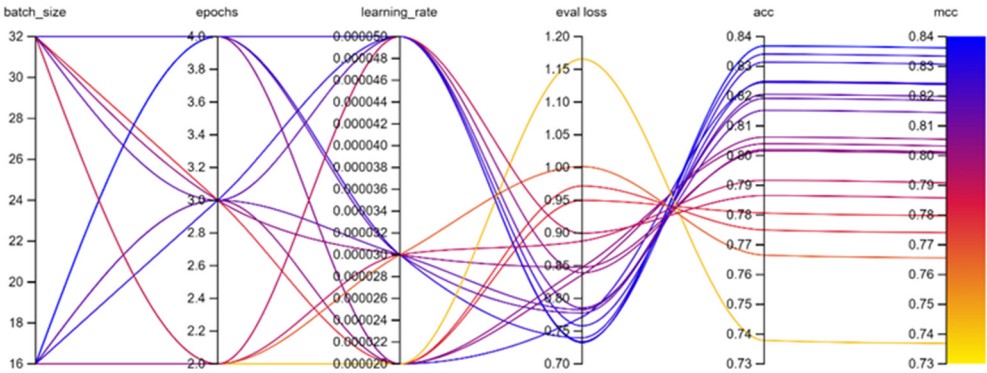

**Figure 7.** Multilingual BERT hyperparameter tuning.

The experiments performed by the authors on Portuguese BERT were also carried out 18 times, for each possible combination of parameters as suggested in [3]. The results showed that the greatest *MCC* achieved was 0.8491 for a batch size value of 16, 4 epochs, and a learning rate of $5 \times 10^{-5}$. In this empirical comparison, a lower batch size and higher epochs and learning rates has shown to be the best combination for fine tuning a BERT model using this data for both the multilingual and Portuguese models. For the best-case scenario in the Portuguese BERT model experiment the *MCC* presented a value of 0.8491, accuracy reached 0.8497, and cross-entropy loss was 0.6941. Table 4 shows Portuguese BERT results for Chapter 90.

**Table 4.** Portuguese BERT results for Chapter 90.

| Batch Size | Epochs | Learning Rate | Cross-Entropy Loss | Accuracy | MCC |
|:---:|:---:|:---:|:---:|:---:|:---:|
| 16 | 4 | $5 \times 10^{-5}$ | 0.6941 | 0.8497 | 0.8491 |
| 16 | 4 | $3 \times 10^{-5}$ | 0.7047 | 0.8432 | 0.8426 |
| 32 | 4 | $5 \times 10^{-5}$ | 0.6946 | 0.8424 | 0.8418 |
| 16 | 3 | $5 \times 10^{-5}$ | 0.7054 | 0.8392 | 0.8385 |
| 16 | 4 | $2 \times 10^{-5}$ | 0.7414 | 0.8320 | 0.8313 |
| 16 | 3 | $3 \times 10^{-5}$ | 0.7420 | 0.8291 | 0.8284 |
| 32 | 4 | $3 \times 10^{-5}$ | 0.7434 | 0.8288 | 0.8281 |
| 32 | 3 | $5 \times 10^{-5}$ | 0.7358 | 0.8281 | 0.8274 |
| 16 | 2 | $5 \times 10^{-5}$ | 0.7707 | 0.8184 | 0.8176 |
| 16 | 3 | $2 \times 10^{-5}$ | 0.8001 | 0.8150 | 0.8142 |
| 32 | 3 | $3 \times 10^{-5}$ | 0.8079 | 0.8114 | 0.8106 |
| 32 | 4 | $2 \times 10^{-5}$ | 0.8165 | 0.8106 | 0.8098 |
| 16 | 2 | $3 \times 10^{-5}$ | 0.8331 | 0.8046 | 0.8038 |
| 32 | 2 | $5 \times 10^{-5}$ | 0.8308 | 0.8039 | 0.8032 |
| 16 | 2 | $2 \times 10^{-5}$ | 0.9309 | 0.7847 | 0.7839 |
| 32 | 2 | $3 \times 10^{-5}$ | 0.9481 | 0.7790 | 0.7782 |
| 32 | 2 | $2 \times 10^{-5}$ | 1.1177 | 0.7459 | 0.7449 |

Results from the 18 different scenarios show that the Portuguese BERT outperforms the multilingual BERT *MCC*. One of the reasons behind this is that Portuguese is a high-resource language to which the specific language model fits better than the BERT multilingual model. Additionally, since all the training data is in Portuguese the BERT Portuguese tends to adapt better to the specific details of the language, getting the inner details more frequently

than models that can handle multiple languages at a time. In this case, the model also improves results when comparing to [16] accuracy results for Chapter 90 with no duplicates and [15] results for the same chapter and scenario 90 but for harmonized system (HS). Figure 8 also presents the hyperparameter tuning process illustrated using weight and biases in [24] for the Portuguese BERT model experiment.

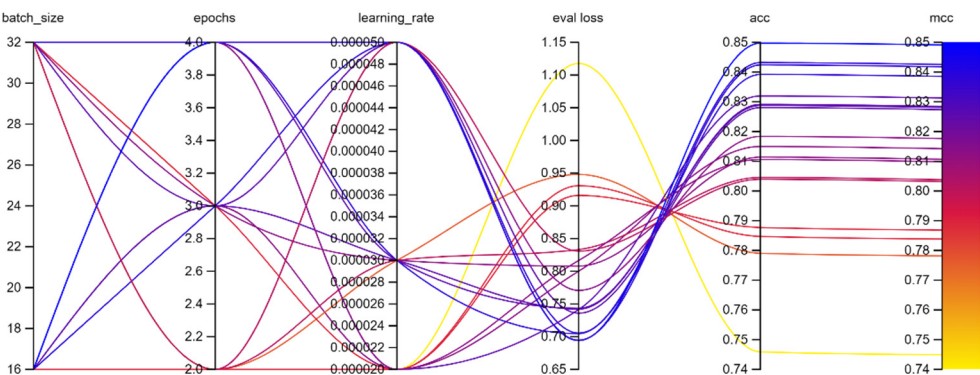

**Figure 8.** Portuguese BERT hyperparameter tuning.

Hence, in both Portuguese and multilingual BERT, the best result was achieved with the smallest batch size between the two options, which corresponds to the number of samples processed before updating the model. Additionally, the smallest learning rate between the three options has shown to be a better hyperparameter option than the others, and finally the number of epochs, which corresponds to the number of times that all the data goes into the model, has been shown to be the greatest possible between the three. Additionally, when considering the 18 scenarios, for both models the *MCC* fluctuated approximately 13% between the worst and the best value, confirming the importance of hyperparameter tuning to get better results.

Having presented the results, mainly about the Matthews correlation coefficient (MCC), it was noticed that it is now possible to train a relevant performance classifier with a low computational cost, mainly due to the knowledge transfer process that the BERT models allow. Thus, with the process of fine-tuning for the classification of product descriptions in their respective MCN codes, it was successfully executed leading to results such as 0.8491 for the *MCC* using the Portuguese pre-trained model in a task with 325 distinct classes. This pretraining process which has a high cost of training allows BERT models to have a relevant performance when compared to other existing traditional methods.

Finally, since both models *MCC*'s result highlighting the relevance of the classifier and as the transformer-based methods work almost like a 'black box' regarding what really led to the models' decision, the authors decided to apply the LIME (local interpretable model-agnostic explanations) proposed in [13] to the Portuguese BERT model classifier. According to its authors, LIME is an explanation technique of any classifier (independent of its inner workings) which focus on presenting an interpretable model local around one prediction in which a human could easily understand and confirm the classifier prediction trust.

The Portuguese BERT classifier was selected to apply the LIME method since it outperformed the multilingual BERT in all proposed scenarios. Using the LIME package and loading the saved trained Portuguese BERT model, a new product description was collected from Siscori import data from November 2021, since the model was trained on data from January 2019 through August 2021 and an unseen instance should be selected to test the classifier performance. Once the simple transformers library required all classes to be integers from 0 to n and all MCN codes were replaced by a respective integer, all prediction probabilities refer to integers numbers represents each MCN code class.

The selected instance (product description) had its correct class as the MCN code 90.26.80.00. The chapter and position part, 90.26, stands for "instruments for measuring or controlling the flow, level, pressure or other variable characteristics of liquids or gases".

Regarding its sub-position, item, and sub-item, to make the entire MCN code, it refers to other instruments that are not classified in controlling the flow, level or pressure specifically. Figure 9 shows the LIME explanation, in which the predicted class (with the highest prediction probability) is the integer 212 with 92%, which corresponds (taking into account the code to integer transformation) to the 90.26.80.00 class, which is the right one for this instance of product description. Other probabilities are spread in other classes, but the prediction is clear enough on the selected class.

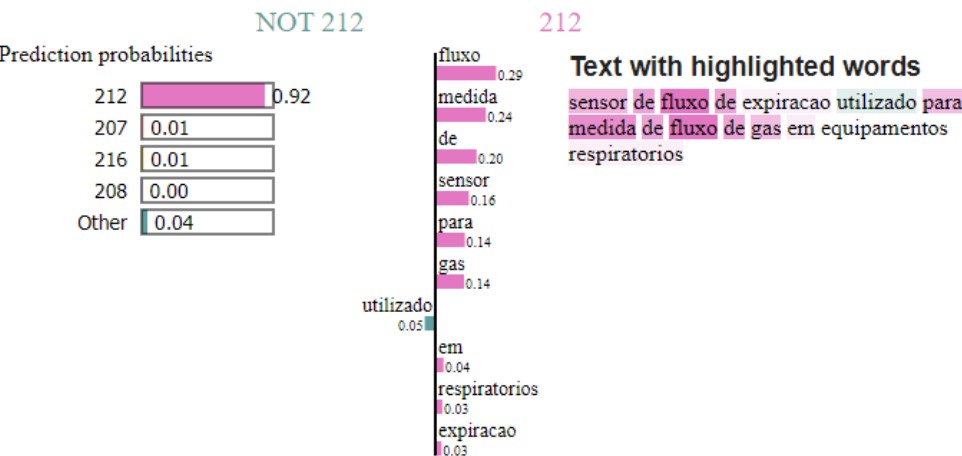

**Figure 9.** LIME applied to BERT Portuguese classifier.

Regarding the highlighted words in pink in Figure 9 that led the classifier to decide for class 212 (90268000), are the words in Portuguese for "flow", "measure", "of", "sensor", "for", "gas", "on", "respiratory", and "expiration" in this order, opposing to "used". All of the mentioned words that led the model to this class selection are words totally related to this specific MCN code in which the instance was classified. The words "flow" and "measure", for example, are totally related to the chapter and position 90.26 and represents key words to distinguish between this position and others, while "respiratory" and "expiration" are probably the words used to distinguish between other sub-position, item, and sub-Item within the same position. LIME method provided an intuitive and interpretable sample in which the authors were able to infer based on what the BERT Portuguese model selected the class, and the prediction has proven to be suitable since the words that were more important to decision-making were truly related to the MCN code and could easily be used to distinguish it from other codes.

Concerning research limitations, since the training process for the BERT models' classifiers are based on real import data, only products that were imported at least once will be part of the training process. This means that if a product with a specific MCN code never had a product imported to Brazil, it will not be among the training data which will make the classifier inadequate to predict those new products. Additionally, despite the Siscori website providing data from 2016 through 2021, not all of these data are suitable for use, since MCN codes are deprecated occasionally which would need extra cleaning and preprocessing when leading with data from early years such as 2016, since some MCN codes would not be active anymore, needing to be removed or have their code updated.

## 5. Conclusions

Given the difficulties in the classification of goods in Brazil and the expensive fines for classification errors, the use of the developed classifier as a starting point in this classification process is obviously an important and relevant result. Considering that Chapter 90 type goods have one of the greatest (classification) errors in the literature related to foreign trade, the classifier can be useful for supporting foreign trade analysts' decisions when classifying goods, especially for goods and products that require more technical knowledge and details.

The trained classifier could also be used by authorities such as the Brazilian Revenue Service. For instance, during the process of customs checking the goods and products descriptions and associated MCN codes are verified. In this case, the classifier could complement, optimize, and improve the existing process, helping to predict what MCN code the good/product should be assigned to, allowing to compare the predicted code with the one provided from customers in the import declaration.

Regarding the model comparisons, as BERT multilingual documentation stated, specific-language methods tend to have better results particularly in feature-rich languages like Portuguese, which is clearly proven in the empirical comparison made in this paper. For the dataset and classification problem, which is the focus of this work, BERT Portuguese has been shown to be the better one between the two models. The Portuguese BERT outperformed multilingual BERT, meanwhile the results with multilingual BERT are encouraging considering the training in 102 languages and even outperformed related work on the same scenarios.

The knowledge transfer process in models such as BERT combined with the availability of import and export records by the Brazilian Revenue Service has made the development of classifiers like this possible. The possibility of fine-tuning the model, as well as its parallel nature, allow for a reduction in training time and make it feasible to run it on local machines or notebooks running in the cloud as shown in this research. In addition to this, the availability of open-source libraries and models allows the sharing of knowledge and the implementation of solutions using state-of-the-art technologies by developers worldwide, as is the case of the multilingual BERT proposed in [3] and Portuguese BERT proposed by [4].

For future work, the authors suggest the training of the Portuguese BERT model on other chapters with the aim of developing an 'upper level' classifier that is able to determine to which chapter a product description belongs to, so it can further be pipelined with chapter-specific classifiers. The Brazilian Revenue Service provides data from 2016 through 2021 from all MCN 96 chapters; thus, by extracting only the first two digits from each sample, it would be possible to train a classifier to distinguish between these 96 classes (chapters). This 'upper level' classifier would perform what a foreign trade analyst does when classifying into which chapter the good fits and the specific chapter classifier would then provide what a chapter specialist does by selecting the appropriate MCN code for the good.

As well as the development of the chapters classifier, the authors suggest the training of a classifier for each chapter, starting from the chapters that are most prone to errors, as reported in the literature. This could lead to a major improvement in foreign trade analysts' daily work routine as done with Chapter 90 in this work. Since there are enough data available and relevant number of classes for each chapter, keeping a classifier specialized for each chapter would allow a more precise prediction on each specific subject.

Additionally, this work used both multilingual BERT and Portuguese BERT on its base form, with 12 layers, due to a reduction in computational cost when dealing with the base form since hyperparameter tuning and cross-validation highly increase the time to run the experiments. Reference [4] also provides a Portuguese BERT model on its large form, with 24 layers, which according to their evaluation benchmarks proved to outperform the base form on all tasks. Considering the increase in classifier performance using the large form, the authors also suggest an experiment with Portuguese BERT large for future works on Chapter 90, as well as other relevant chapters for the industry.

In addition to the public data provided by the Brazilian Revenue Service and the explanation regarding each MCN code composition, [6] also provides "Harmonized System Explanatory Notes" (HSENs) for each chapter and position. These notes explain in detail its scope as well as legal notes that help understand which goods do not belong to the chapter. The authors suggest research on different models or approaches that could take those notes into account and that could deal with these explanatory notes on what the chapters refer

to. Since all these notes have a detailed explanation on each part, a search for a model that could take this written detailed explanation as input would be relevant for the field.

Finally, after the proposal of the transformers in [2] and BERT in [3], several other models were presented such as RoBERTa in [25] and DistilBERT in [26] which propose some changes from the original BERT model and could potentially have improved results or performance when used to build and train classifiers when compared to the ones presented in this work. The authors also are considering researching what the specific changes on those models are and their application to the Siscori data in order to train a classifier, as done in this paper to the multilanguage and Portuguese BERT models.

**Author Contributions:** Conceptualization, R.R.d.L. and J.R.B.; methodology, R.R.d.L. and A.M.R.F.; validation, R.R.d.L. and A.M.R.F.; formal analysis, R.R.d.L. and A.M.R.F.; investigation, R.R.d.L., B.A.d.S. and A.M.R.F.; data curation, R.R.d.L.; writing—original draft preparation, R.R.d.L. and A.M.R.F.; writing—review and editing, P.C. and V.R.Q.L.; visualization, R.R.d.L., A.M.R.F., P.C. and V.R.Q.L.; supervision, A.M.R.F.; project administration, A.M.R.F.; funding acquisition, P.C. and V.R.Q.L. All authors have read and agreed to the published version of the manuscript.

**Funding:** This work was supported by national funds through the Foundation for Science and Technology, I.P. (Portuguese Foundation for Science and Technology) by the project UIDB/05064/2020 (VALORIZA—Research Center for Endogenous Resource Valorization), and Project UIDB/04111/2020, ILIND–Lusophone Institute of Investigation and Development, under project COFAC/ILIND/COPEL ABS/3/2020. Also, this work is funded by FCT/MCTES through national funds and when applicable co-funded EU funds under the project UIDB/50008/2020.

**Institutional Review Board Statement:** Not applicable.

**Informed Consent Statement:** Not applicable.

**Data Availability Statement:** Data presented in this paper are open data by the Brazilian Government and could be accessed until 16 December 2021 through the Siscori Website (https://siscori.receita.fazenda.gov.br/apoiosiscori/) and after that date the data can be accessed on Stat Comex Website (http://comexstat.mdic.gov.br/pt/geral). All data referred to foreign trade in Brazil are available on the Foreign Trade Statistics in Open Data website (https://www.gov.br/produtividade-e-comercio-exterior/pt-br/assuntos/comercio-exterior/estatisticas/base-de-dados-bruta) which is also under the Attribution-NoDerivs 3.0 Unported (CC BY-ND 3.0) as referred in page footer.

**Conflicts of Interest:** The authors declare no conflict of interest.

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
