# Peer review of "An Empirical Comparison of Portuguese and Multilingual BERT Models for Auto-Classification of NCM Codes in International Trade"

_2504-2289, doi:10.3390/bdcc6010008_

Round 1

Reviewer 1 Report

For me it is a sound and important conclusion that BERT adapted to Portuguese is better than Multilingual BERT supported by the empirical study.
The only thing I would suggest is to discuss why the authors used these types of models and what are their lackings as well as their benefits for the readers.
BERT is a well known and used method based by Google. The authors used still enough data to train the model (265818 in 325 classes).
In addition, they used hyperparameter tuning to optimize the learning effects (by k-fold cross-validation training). MCC remained consistent for binary and mutliclass classifiers.
What should be discussed in detail is the significance in the dfference between MCCbertportugese=0.8497 and MCCbertmultilingual=0.8362. 
Please also remove in Line 437: "1 Tables may have a footer."

Finally, the authors should explain the benefit of their study to the audience and the benefit of BERTportuguese for their users.

Author Response

Dear reviewer, We greatly appreciate your availability to review our article, we send you a letter detailing all the changes made.

Reviewer 2 Report

--would be better the authors  apply the LIME /SHAP  to the prosed method.

Local Interpretable Model-Agnostic Explanations (lime)

The introduction should be rewritten to clarify its message. Drawbacks of former proposals should be clearly indicated and innovations and new ideas highlighted.

-Your abstract does not highlight the specifics of your research or findings.

-Problem statement, objective are not clear. The authors should give a more accurately description about the existing methods.

-In the related work section, the authors have added short comments to new references; among these, more details of the method proposed in the above cited paper have to be reported. Should be better explained to outline the limitations of the approaches.

-The paper's presentation should be improved

-I feel that more explanation would be need on how the proposed method is performed

-If no one has proposed before a method like the proposed algorithm, this claim should be highlighted much more. Else, it should be indicated who has done this, and it should be indicated what the innovations of the current paper are.

- The proposal does not present information so that the application can be reproduced by readers.

- Improve text formatting.

-the authors did not compare the proposed method with existing method, Btw, clearly explain how do authors re-examined the mentioned approaches in order to use them in a fair comparison with the proposed algorithm. The authors also must describe why the proposed outperformed other existing method

-Tables & figures present a lot of statistics but needs more detailed explanation (which is missing in the text).

-Please, also provide a paragraph with three to five clear positive impacts of your algorithm.

- the authors should further detail the preparation of the dataset.

-the figures must be in better format & resolution

-the machine/deep learning methods must explain in details. How to set the parameters.

-There are no real insightful conclusions drawn from the study and no suggestions for practical use of the results. Therefore, the conclusion section should be totally rewritten in order to:

- You must more clearly highlight the theoretical and practical implications of your research

-Discuss research contributions.

-Indicate practical advantage, discuss research limitations, and supply 2-3 solid and insightful future research suggestions.

Author Response

(The authors gave the same response as above.)
